# A Systematic Review of Oxygen Therapy for the Management of Medication-Related Osteonecrosis of the Jaw (MRONJ)

**Roberto Sacco [1,2,3,]*, Racheal Leeson [4], Joseph Nissan [5], Sergio Olate [6],**
**Carlos Henrique Bettoni Cruz de Castro [7], Alessandro Acocella [8] and Anand Lalli [9]**

[1] Oral Surgery Department, Barts and the London School of Medicine and Dentistry, London E1 2AT, UK
[2] CPD Department, Eastman Dental Institute–University College of London, London WC1X 8LT, UK
[3] Oral Surgery Department, King's College Hospital, London SE5 9RW, UK
[4] Oral Surgery Department, Eastman Dental Institute–University College of London, London WC1X 8LD, UK; r.leeson@ucl.ac.uk
[5] Oral Rehabilitation Department, School of Dentistry, Tel-Aviv University, Tel-Aviv 69978, Israel; nissandr@post.tau.ac.il
[6] Oral and Maxillofacial Surgery Department, Universidad de La Frontera, Temuco 1145, Chile; sergio.olate@ufrontera.cl
[7] Oral Surgery Department, Pontifícia Universidade Católica de Minas Gerais, Belo Horizonte 30535-360, Barzil; carlosbettoni1@yahoo.com.br
[8] Private Practice, Florence 50129, Italy; alessandroacocella@yahoo.it
[9] Centre for Oral Immunobiology and Regenerative Medicine, Barts and the London School of Medicine and Dentistry, London E1 4NS, UK; a.lalli@qmul.ac.uk
*   Correspondence: r.sacco@ucl.ac.uk

**Abstract:** Background: Medication-related osteonecrosis of the jaw (MRONJ) can be a life changing iatrogenic complication of antiresorptive and antiangiogenic drug therapy. It is most often associated with high doses of these medications that are used to prevent skeletal-related events in patients with cancer and bone pathologies. Unfortunately, managing MRONJ lesions has proven difficult and remains a major challenge for clinicians. Due to the lack of efficacy in treating MRONJ by surgical modalities (local debridement and free flap reconstruction), the nonsurgical management of MRONJ is still advocated to aid healing or avoid disease progression. The aim of this systematic review is to identify, analyse and understand the published evidence related to the success of oxygen therapies such as ozone (OT) and hyperbaric oxygen (HBO) in treating MRONJ. Material and methods: A multi-database (PubMed, MEDLINE, EMBASE, CINAHL and Cochrane CENTRAL) systematic search was performed by three authors. The identified articles were independently assessed for their risk of bias. Any type of study evaluating humans treated with antiresorptive and antiangiogenic drugs were considered. The aim is primarily to evaluate the success of OT and HBO in resolving MRONJ and secondarily to identify any improvements in quality of life (QoL), rate of complications, time-to-event and severity of side effects related to these treatments. Results: In total, just 13 studies were eligible for analysis. A pooled total of 313 patients (HBO group n = 82; OT group n = 231) described in these studies have shown good tolerance for oxygen therapies. Complete resolution of MRONJ was reported in 44.58% of OT patients but only 5.17% of the HBO group. Progression of MRONJ was reported only in the HBO studies in 10.34% of cases (6 patients). The quality of evidence was low or very low in all studies. This was due to limitations in how the studies were designed, run and reported. Conclusions: Based on the limited data available, it is difficult to suggest OT is better or worse than HBO or whether it is better than a placebo. As the level of evidence available is low, this necessitates larger well-designed trials to justify these interventions for patients affected by MRONJ.

**Keywords:** osteonecrosis; bisphosphonate-related osteonecrosis of the jaw; antiresorptive drugs; antiangiogenic drugs; ozone therapy; hyperbaric oxygen therapy

## 1. Introduction

The term 'medication-related osteonecrosis of the jaws' (MRONJ) refers to a potentially serious iatrogenic complication of treatment with medications, such as antiangiogenic or antiresorptive drugs. These drug families are used primarily for the treatment of malignancies (e.g., multiple myeloma or bone metastases) but also have important roles in the management of osteoporosis, Paget's disease and hypercalcemia. The bone targeting agents (BTAs) can reduce the risk of skeletal-related adverse events to protect the skeleton in patients with primary or secondary bone pathology [1,2].

Since the first reports of bisphosphonate-related osteonecrosis of the jaw in 2003, an increasing number of reports have been published showing similar clinical complications associated with other drugs. New evidence has shown that along with bisphosphonates (BPs), other BTAs such as denosumab also cause osteonecrosis of the jaw bones. In addition, monoclonal antibodies able to bind and selectively inhibit vascular endothelial growth factor-A (VEGF-A), specifically mammalian target of rapamycin (mTOR) inhibitors, can also cause similar lesions [3–5]. For this reason, the term MRONJ was adopted in the 2014 position paper of the American Association of Oral and Maxillofacial Surgeons (AAOMS) [3]. The medications currently reported to be associated with MRONJ are listed in Table 1 (antiresorptive drugs) and Table 2 (antiangiogenic agents) [6,7].

The AAOMS position paper states that "patients may be considered to have MRONJ if all the following characteristics are present: current or previous treatment with antiresorptive or antiangiogenic agents; exposed bone or bone that can be probed through an intraoral or extraoral fistula in the maxillofacial region that has persisted for longer than 8 weeks; and no history of radiation therapy to the jaws or obvious metastatic disease to the jaws". However, it is accepted that some patients may present with non-specific symptoms and may not have evidence of exposed bone such as those classified as Stage 0 by the AAOMS staging system [8]. The term 'stage 0' was first used by Mawardi et al. to gather suspected MRONJ cases presenting with clinical and radiological signs of disease other than intraoral bone exposure [9]. The AAOMS classification and staging system is based entirely on the intraoral presentation, which they propose should also guide potential treatment. This has been criticised in studies where it was highlighted that radiological findings are an important part of the clinical picture necessary for accurate MRONJ staging (Tables 3 and 4) [6,8,10].

The major risk factor for the development of MRONJ is dento-alveolar surgery, with a history of tooth extraction or oral surgery procedure (apicectomy or cystectomy) reported in 52% to 80% of patients with MRONJ [11–13]. The overall risk of developing MRONJ after dento-alveolar surgery, in patients on IV bisphosphonates ranges from 1.6% to 14.8% in comparison to 0.5% for patients taking oral bisphosphonates [6].

To date, there is no current standard for the treatment of MRONJ associated with antiresorptive or antiangiogenic drug therapy. Several treatment options have been described, with the earlier stages of MRONJ reportedly responding well to conservative management such as topical or systemic antibiotics, or limited bone debridement, although this still remains controversial [14,15].

Ozone therapy (OT) and hyperbaric oxygen therapy (HBO) have been reported as effective adjunctive therapies in situations where normal bony wound healing is impaired, such as osteoradionecrosis and chronic osteomyelitis of the jaw [16–18].

**Table 1.** Antiresorptive drugs associated with medication-related osteonecrosis of the jaws (MRONJ). Btl: Bottle; IM: Intramuscular; IV: Intravenous; MM: Multiple myeloma; PO: Orally; SC: Subcutaneous; SRE: Skeletal-related event; Tab: Tablet.

| Pharmacologic Active Ingredient | Formulation | Route of Administration | Indication and Frequency |
|---|---|---|---|
| Alendronic acid (sodium salt) | Tab 70 mg<br>Tab 10 mg | PO | Treatment of postmenopausal osteoporosis (70 mg/week)<br>Treatment of osteoporosis in men (70 mg/week)<br>Treatment and prevention of osteoporosis induced by glucocorticoids (70 mg/week) |
| Alendronic acid + cholecalciferol | Tab 70 mg/5600 UI | PO | Treatment of postmenopausal osteoporosis in patients with unsupplemented vitamin D deficit (70 mg/week) |
| Ibandronic acid (monosodium salt monohydrate) | Tab 50 mg<br>Btl 6 mg/6 mL<br>Tab 150 mg<br>Btl 3 mg/3 mL | PO<br>IV<br>PO<br>IV | Prevention of SREs in breast cancer patients with bone metastases (50 mg/day p.o. or 6 mg every 3–4 weeks iv.)<br>Treatment of hypercalcemia of malignancy<br>Treatment of postmenopausal osteoporosis in patients at high risk of fracture (150 mg/4 weeks p.o. or 3 mg every 3 months iv.) |
| Neridronate acid (sodium salt) | Btl 25 mg/2 mL<br>Btl 100 mg/8 mL | IV/IM.<br>IV | Osteogenesis imperfecta (2 mg/kg/3 months)<br>Treatment of Paget's disease (different schedules) |
| Risedronic acid | Tab 35 mg<br>Tab 5 mg | PO | Treatment of postmenopausal osteoporosis (35 mg weekly or 5 mg daily)<br>Treatment and prevention of osteoporosis induced by glucocorticoids (35 mg weekly or 5 mg daily)<br>Treatment of Paget's disease |
| Zoledronic acid (monohydrate) | Btl 4 mg/5 mL<br>Btl 5 mg/100 mL | IV<br>IV | Prevention of SREs in cancer patients with bone metastases or MM (4 mg every 3–4 weeks). Treatment of hypercalcemia of malignancy<br>Treatment of osteoporosis in postmenopausal women, in men at increased risk of fracture, including those with a recent hip fracture from minor trauma (5 mg once per year)<br>Treatment of Paget's disease |
| Denosumab | Btl 120 mg<br>Btl 60 mg | SC<br>SC | Prevention of SREs in cancer patients with bone metastases (120 mg every 4 weeks)<br>Treatment of hypercalcemia of malignancy.<br>Osteoporosis (60 mg sc. every 6 months) |

**Table 2.** Antiangiogenic drugs associated with MRONJ. IV: Intravenous; MM: Multiple myeloma; PO: Orally; SC: Subcutaneous; Btl: Bottle; Tab: Tablet.

| Pharmacologic Active Ingredient | Formulation | Route of Administration | Indication and Frequency |
|---|---|---|---|
| Bevacizumab | Btl 400 mg Btl 100 mg | IV | Metastatic breast cancer (10 mg/kg every 2 weeks or 15 mg/kg every 3 weeks); colorectal cancer (5 mg/kg or 10 mg/kg every 2 weeks); lung/ovarian cancer (7.5 mg/kg or 15 mg/kg every 3 weeks); renal cell cancer (10 mg/kg every 2 weeks); glioblastoma (10 mg/kg every 2 weeks) |
| Sunitinib | Tab 12.5 mg | PO | Renal cell cancer, GISTs and neuroendocrine tumours (50 mg/day for 4 weeks) |
| Sorafenib | Tab 200 mg | PO | Renal cell cancer (800 mg/day) |
| Pazopanib | Tab 200 mg Tab 400 mg | PO | Renal cell cancer (200–800 mg/day) |
| Thalidomide | Tab 50 mg | PO | Myeloma (400 mg/day for 6 weeks) |
| Lenalidomide | Tab 5, 10, 15 and 25 mg | PO | Myeloma (tailored doses) |
| Everolimus | Tab 5 and 10 mg | PO | Renal cell cancer, breast cancer (10 mg every day) |
| Temsirolimus | Btl 30 mg | IV | Renal cell cancer (25 mg every week) |

**Table 3.** MRONJ staging from the American Association of Oral and Maxillofacial Surgeons (AAOMS) position paper (2014) [6].

| Stage | MRONJ Clinical Findings |
|---|---|
| At risk category | No apparent necrotic bone in patients who have been treated with either oral or IV bisphosphonates |
| Stage 0 | No clinical evidence of necrotic bone, but non-specific clinical findings, radiographic changes and symptoms |
| Stage 1 | Exposed and necrotic bone, or fistulae that probes to bone, in patients who are asymptomatic and have no evidence of infection |
| Stage 2 | Exposed and necrotic bone, or fistulae that probes to bone, associated with infection as evidenced by pain and erythema in the region of the exposed bone with or without purulent drainage |
| Stage 3 | Exposed and necrotic bone or a fistula that probes to bone in patients with pain, infection, and one or more of the following: exposed and necrotic bone extending beyond the region of alveolar bone,(i.e., inferior border and ramus in the mandible, maxillary sinus and zygoma in the maxilla) resulting in pathologic fracture, extra-oral fistula, oral antral/oral nasal communication or osteolysis extending to the inferior border of the mandible of sinus floor |

**Table 4.** MRONJ staging (clinical and radiological findings) proposed by Campisi et al. 2011 [10].

| Stage | MRONJ Features |
|---|---|
| Stage 1 | Focal ONJ<br>Clinical signs and symptoms: bone exposure, sudden dental mobility, non-healing post-extraction socket, mucosal fistula, swelling, abscess formation, trismus and gross mandible deformity hypoesthesia/paraesthesia of the lips CT signs: increased bone density limited to the alveolar bone region (trabecular thickening and focal osteosclerosis), with or without the following signs: markedly thickened and sclerotic lamina dura, persisting alveolar socket and cortical disruption |
| Stage 2 | Diffuse ONJ<br>Clinical signs and symptoms: same as stage 1<br>CT signs: increased bone density extended to the basal bone (diffuse osteosclerosis), with or without the following signs: prominence of the inferior alveolar nerve canal, periosteal reaction, sinusitis, sequestra formation and oro-antral fistula |
| Stage 3 | Complicated ONJ<br>Same as stage 2, with one or more of the following:<br>Clinical signs and symptoms: extra-oral fistula, displaced mandibular stumps and nasal leakage of fluids<br>CT signs: osteosclerosis of adjacent bones (zygoma and hard palate), pathologic mandibular fracture and osteolysis extending to the sinus floor |

HBO increases local concentrations of reactive nitrogen species (RNS) and reactive oxygen species (ROS) by providing substrates (oxygen and L-arginine) for nitric oxide synthase, as well as by the generation of superoxide [19,20]. ROS and RNS influence osteoclast (OC) differentiation and activity and participate in the regulation of various aspects of bone metabolism [21–23]. Nitric oxide (NO) is constitutively synthesised by both OCs and osteoblasts (OBs) and has contrasting biphasic effects. At lower levels, NO decreases bone resorption and stimulates its turnover; whereas at higher concentrations, NO promotes inflammatory processes and inhibits bone formation [24]. Indeed, ROS stimulate the expression of receptor activator of nuclear factor kappa-B ligand (RANKL), changing the RANKL/osteoprotegerin ratio and favouring OC differentiation, and avoiding osteopetrosis in animal models [22,25]. Authors have also suggested that HBO-generated ROS could induce suppression of OC activity and promote bone healing [21,23]. Recent research has shown that HBO-generated ROS and RNS induce stem cell mobilisation, vasculogenesis, mitochondrial biogenesis and preconditioning [26–29].

Ozone dissolves physically in biological water (physiological saline, plasma, lymph, urine). All these reagents act as donor electrons and are oxidisable, and participate in the ozonation process and the consequent formation of ROSs and lipid oxidation products (LOPs). These molecules are responsible for the biochemical actions of ozone and function as biochemical regulators of inflammation at distinct times and physiological concentrations [30,31]. Through interactions with cellular components and depending on the concentration of ozone in the tissue, these molecules can trigger biological effects that are either therapeutic or detrimental to health [30,31].

During normal metabolism, osteoclasts, platelets, lymphocytes, neutrophils, monocytes and fibroblasts can induce the formation of ROS [32,33]. When in excess, these ROSs can trigger damage to cellular constituents, extracellular components and affect the metabolism of the cells responsible for extracellular matrix synthesis—fundamental in tissue repair—which leads to apoptosis and cellular necrosis [34,35].

The aim of this review is to analyse all available evidence and evaluate the reported outcome of oxygen therapy as treatment for patients affected by MRONJ.

## 2. Material and Methods

This systematic review was performed according to Preferred Reporting Items for Systematic Reviews and Meta-Analyses (PRISMA) guidelines [36]. The following databases were interrogated: PubMed, MEDLINE, EMBASE, CINAHL and the Cochrane Central Register of Controlled Trials (CENTRAL). A three-stage screening approach was used to ensure precision and the quality of the search. The screening of titles and abstracts was carried out independently by three authors (RS, RL and SO) to eliminate any irrelevant material (i.e., reviews, animal studies, non-clinical studies and studies that did not report patients undergoing oxygen therapy treatments). Disagreements were resolved by discussion until a consensus was reached.

A data screening and abstraction form was used to:

(1) Verify the study eligibility derived from the inclusion/exclusion criteria.
(2) Carry out the methodological quality assessment.
(3) Extract data on study characteristics and outcomes for the included studies.

The authors of any studies eligible for inclusion in the review, unless without sufficient information, were contacted directly (Figure 1).

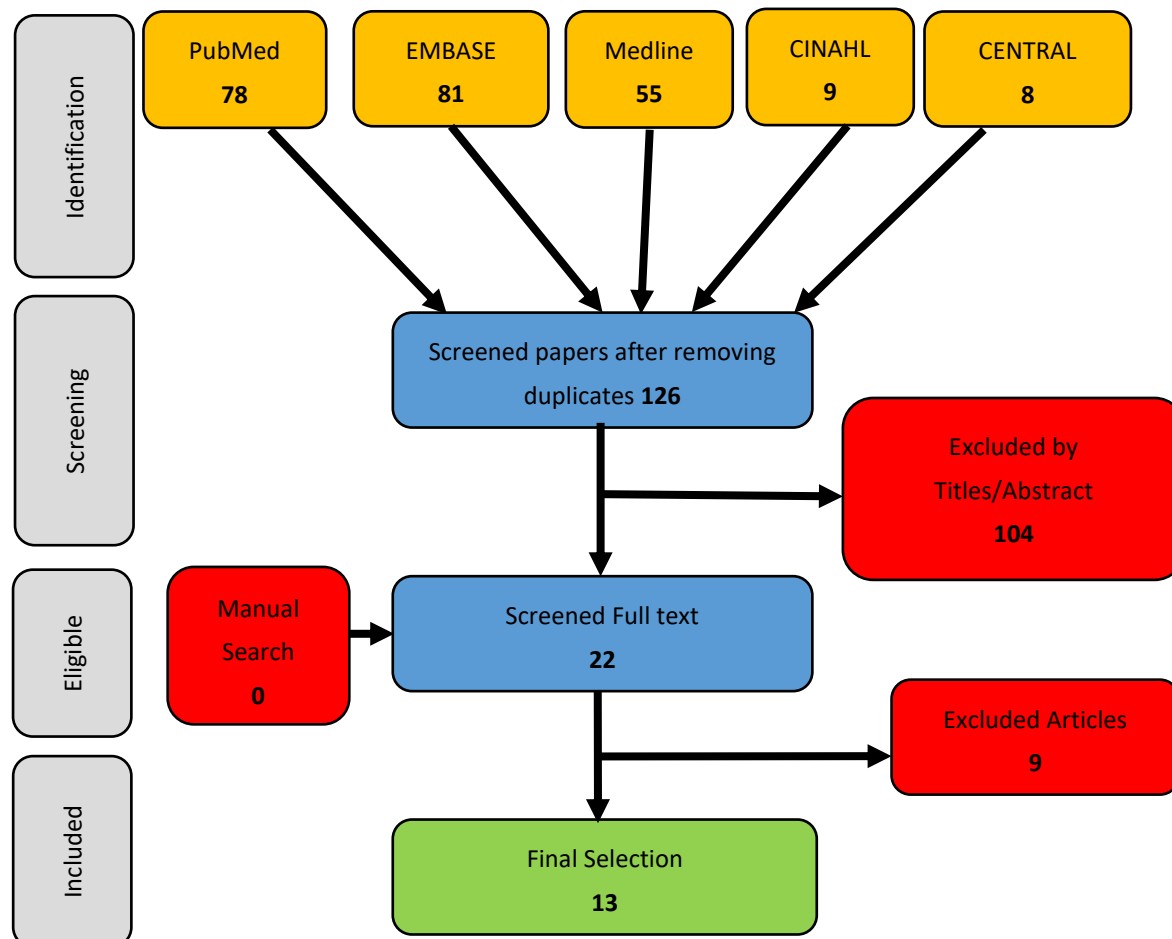

**Figure 1.** Study flow diagram.

## 3. Criteria for Inclusion in this Review

### 3.1. Types of Studies

The types of studies included in the research strategy were published or unpublished randomised controlled trials, case-controlled trials, case series, retrospective studies and case reports. Papers were obtained from January 2003 to September 2018. Animal studies, reviews and those studies including patients with a previous history of radiation therapy to the head and neck regions were excluded. No language restrictions were imposed to the search.

### 3.2. Types of Participants

The review considered studies involving patients who developed MRONJ and subsequently underwent OT and/or HBO treatment. No restriction of age, gender or ethnic origin was applied. There was also no restriction on the minimum number of patients included in the studies.

### 3.3. Types of Interventions

Patients affected by MRONJ who underwent OT and/or HBO as either standalone or adjuvant treatment were considered.

### 3.4. Objectives

The objectives was to assess the therapeutic effects of HBO and OT in patients exposed to antiresorptive or antiangiogenic drugs and affected by MRONJ. Moreover, it was to assess the effects of OT and HBO therapy as standalone or adjuvant treatments (either singly or in combination to other treatments) in people with manifest MRONJ.

## 4. Types of Outcome Measures

➢ **Primary outcomes**

- Healing of MRONJ as indicated by one or more of the following indicators:

  ○ Improvement in the clinical grade of the lesions according to the AAOMS staging of MRONJ (Table 3).
  ○ Wound healing (yes or no).
  ○ Plain film radiological examination (improvement of sclerotic changes, mottling and bone fragmentation, improvement of formed sequestrum or persistent extraction sockets), computed tomography (CT) scan, magnetic resonance imaging (MRI) (surface area of the bone disease, localisation, evidence of bone marrow disease), positron emission tomography (PET)/CT imaging (decreased abnormal focal uptake) [37].
  ○ Healing of sinus tract or deep periodontal pockets.
- Rate of progression of MRONJ

➢ **Secondary outcomes**

- Quality of life (QoL).
- Time-to-event.
- Rate of complications and side effects of the intervention.

For the 'complications' outcome measure, interventions involving an interruption or delay of antiresorptive or antiangiogenic treatments, or progression of the underlying disease (e.g., fracture in osteoporosis or disease progression in cancer), were considered to be complications of the intervention.

For QoL measures, we reported whether validated scales were used. Non-validated scales were not excluded a priori. QoL had to have been measured at baseline and at least once during follow-up.

## 5. Data Extracted

Data extracted from the studies included number of patients; patient gender and age; predisposing factors for and localisation of MRONJ; type of antiangiogenic or antiresorptive drugs and their cumulative dose; clinical indications for the drug or combined therapy; type of intervention; complications; follow up time; MRONJ evolution and MRONJ recurrence.

All selected papers were carefully read to identify author(s); year of publication; study design; population and treatment characteristics.

In the case of missing information, we contacted the authors and allowed six weeks for a reply. If the information was still missing, we then indicated the missing data as 'Not Reported (NR)' in the text and tables.

## 6. Review Quality Assessment Data

Two review authors (RS, AL) appraised the risk of bias in the included study with the tool recommended by the Cochrane Handbook for Systematic Reviews of Interventions as appropriate for randomised control trials (RCTs) [38]. Moreover, the authors used the CARE Checklist for case report and the Modified Delphi Checklist for the case series studies [39,40]. We referred instances of disagreement in risk of bias assessments to one of the other members of the review team (SO) and resolved them by discussion.

## 7. Results

A total of 13 articles were included in this review. Of these, 6 reported patients treated with HBO and 7 articles included patients treated using OT. All the published data described patients treated from 2006 to 2015. The types of articles included in this research were case series (n = 9), case reports (n = 3) and RCTs (n = 1) [41–53]. Results were expressed as descriptive statistics because of significant heterogeneity in the published data.

### 7.1. List of Excluded Studies

Currently the treatment of MRONJ is controversial, but many researchers agree that intravenous drug administration or longer period of drug intake contribute to high risk of developing MRONJ [54–56].

We originally considered 22 studies to be potentially eligible for inclusion, but after inspection of the full papers, 9 were excluded for not meeting the inclusion criteria for this review [57–65] (Table 5).

**Table 5.** List of excluded studies.

| Authors | Type of Intervention | Number of Patients | Type of Drug | Type of Study | Outcome |
|---|---|---|---|---|---|
| Petrucci et al. 2007 [57] | OT + Surgery | 12 | IV-BP | Letter to editor | 8 patients (75%) achieved complete resolution of ONJ, and 4 (25%) achieved improvement with persistence of lesion |
| Agrillo et al. 2007 [58] | OT prevention strategy for dental extraction | 15 | NR | Case Series | No development of ONJ |
| Yamazaki et al. 2010 [59] | HBO | 1 | Oral BP | Abstract | Improvement with persistence of lesion |
| Karakinaris et al. 2013 [60] | HBO + Drug holiday + Surgery | 25 | Unclear | Poster/Presentation | All patients free from ONJ |
| Salcedo Gil et al. 2013 [61] | Retrospective Comparative (HBO group vs. No-HBO group) as adjuvant therapy | 15 HBO Vs 15 No-HBO | BP | Poster/Presentation | Significant improvement with persistence of lesion and or stabilization |

**Table 5.** *Cont.*

| Authors | Type of Intervention | Number of Patients | Type of Drug | Type of Study | Outcome |
|---|---|---|---|---|---|
| Zaslavskaya et al. 2013 [62] | OT + removal of sequestrum | 30 | IV-BP | Poster/Presentation | All patients free from ONJ |
| Asaka 2014 [63] | HBO + Minimal surgery | 8 | NR | Poster/Presentation | All patients presented good clinical and radiological result |
| Hamada 2014 [64] | HBO + Surgery + stopping BP | 3 | IV-BP and Oral BP | Poster/Presentation | All patients free from ONJ |
| Yahoo et al. 2018 [65] | OT + Surgery | 2 | One on DZB and one on IV-BP | Poster Presentation | All patients free from ONJ |

*7.2. HBO Study Analysis*

In total just 6 articles were included in the analysis of HBO therapy comprising case reports (n = 2), case series (n = 3) and RCTs (n = 1). All studies were published from 2006 to 2015. A total of 82 patients with a mean age 66.3, 45 female (54.88%) and 37 male (45.12%) cases, were treated using different protocols of HBO therapy (Tables 6 and 7). None of the manuscripts reported patients treated with antiangiogenic drugs.

**Table 6.** Studies included in the hyperbaric oxygen (HBO) analysis, including number of patients treated and evidence level. Case series (CS); case report (CR); randomised control clinical trial (RCT); Levels of Evidence for Prognostic Studies Adapted from the American Society of Plastic Surgeons (https://www.plasticsurgery.org/Documents/medical-professionals/health-policy/evidence-practice/ASPS-Rating-Scale-March-2011.pdf).

| Author(s) | Type of Study | Total Number of Patients | Level of Evidence |
|---|---|---|---|
| Shimura et al. 2006 [41] | CR | 1 | Level 5 |
| Lee et al. 2007 [42] | CS | 2 | Level 4 |
| Freiberger et al. 2007 [43] | CS | 16 | Level 4 |
| Lee et al. 2011 [44] | CS | 13 | Level 4 |
| Freiberger et al. 2012 [45] | RCT | 49; (3 patients died at early stage of study) | Level 2 |
| Fatema et al. 2015 [46] | CR | 1 | Level 5 |

The reported indications for antiresorptive drug treatment were multiple myeloma (37.80%), osteoporosis (21.95%), breast cancer (18.29%), prostate cancer (4.88%), non-Hodgkin lymphoma (2.44%), sarcoidosis (1.22%) and macroglobulinaemia (1.22%). In addition, a significant number of patients were logged with no specific indication (n = 10, 12.20%) (Table 8).

The most common site for MRONJ was the mandible (18.29%) followed by the maxilla (3.65%). In 2.43% of patients, MRONJ lesions were reported in both (Table 7). However, in 75.60% the MRONJ site was not reported (NR). The drug most commonly responsible of MRONJ was Zoledronate (12.19%), but only 33 patients out of 82 had this detail reported (40.24%).

HBO was most commonly used as a neoadjuvant and/or adjuvant therapy, followed by surgery in 4 studies out of 6. In one study the HBO therapy was given as standalone treatment (Table 9).

The patients were followed for a period of time ranging from 1 to 32 months. At the end of the follow up, MRONJ was seen as completely resolved in 5.17% of the cases (n = 3), while the majority (48.27%) of the patients (n = 28) were reported to have some benefit due to stability or improvement of the disease presentation. In just 10.34% of patients (n = 6), the disease progressed, but for 13 patients (22.41%), data were not available (Table 10).

**Table 7.** HBO Preoperative epidemiologic analysis (age, sex, predisposing factors and site of the necrosis involved). M: male; F: female; not reported (NR), Standard of Care (StC).

| Study | Patient Numbers | Age/Sex | Triggering Cause | Site of the Necrosis Involved |
|---|---|---|---|---|
| Shimura et al. 2006 [41] | 1 | 60 M | Spontaneous | 1 in mandible |
| Lee et al. 2007 [42] | 2 | 84 F<br>76 M | Dental implant;<br>Bone graft surgery | 1 in the maxilla;<br>1 in the mandible |
| Freiberger et al. 2007 [43] | 16 | 63 F, 69 F, 57 M, 53 M, 70 M, 45 F, 62 M, 59 M, 78 F, 56 M, 52 M, 52 F, 72 M, 77 M, 43 F, 63 M (total patients 6 female and 10 male) | NR | 12 in the mandible;<br>2 in the maxilla;<br>2 in both maxilla and mandible |
| Lee et al. 2011 [44] | 13 | 62 M, 87 M, 54 F, 81 F, 68 M, 75 F, 70 F, 70 F, 57 F, 83 F, 76 M, 74 F, 62 F (total patients 4 male and 9 female) | NR | NR |
| Freiberger et al. 2012 [45] | 27 (StC-group), 22 (HBO-StC group). Total Patients 46 3 early dead during study (2 on the StC StC-group and 1 on the HBO-StC group). | (mean age, 66 years; 57% female) | NR | NR |
| Fatema et al. 2015 [46] | 1 | 80 F | Dental extraction | 1 in the mandible |

**Table 8.** HBO preoperative pharmacological analysis: type of drugs, indication for drug therapy, and time of drug exposure; Minodronate (MI); Zoledronate acid (ZOL); Pamidornate (PAM); Bisphosphonate (BP); Alendronate (ALD); Risedronate (RES); Ibadronate (IBA); Osteoporosis (OP); Multiple Myeloma (MM); Breast Cancer (BC); Sarcoidosis (SC); Prostate Cancer (PC); non-Hodgkin lymphoma (nHL); Unclear—the number expressed do not match the actual sample analysed; Not reported (NR).

| Study | Type of Drug | Indication for Drug Therapy | Time of Drug Exposure |
|---|---|---|---|
| Shimura et al. 2006 [41] | MI | MM × 1 | 32 months |
| Lee et al. 2007 [42] | 2 × ALD | OP × 2 | 1 in more than 108 months; 1 NR |
| Freiberger et al. 2007 [43] | 2 ZOL +PAM<br>7 ZOL<br>6 BP (unknown)<br>1 PAM | BC × 3<br>MM × 10<br>Macroglobulinemia × 1<br>SC × 1<br>PC × 1 | Unclear |
| Lee et al. 2011 [44] | ZOL × 3<br>ALD × 6<br>RES × 3<br>IBA × 1 | OP × 7<br>MM × 1<br>PC × 3<br>nHL × 2 | NR |
| Freiberger et al. 2012 [45] | Unclear | OP (C-group × 5; S-group × 3);<br>MM (C-group × 9; S-group × 10);<br>BC (C-group × 7; S-group × 5);<br>Other indication (C-group × 6; S-group × 4) | Unclear |
| Fatema et al. 2015 [46] | RES | OP | 24 months |

**Table 9.** HBO operative analysis including type of intervention and the stage of the disease. Platelet Rich Plasma (PRP); Standard of Care (StC); Antibiotics (ABX); Mouthwash (MW); American Association of Oral and Maxillofacial Surgery (AAOMS); not applicable (N/A).

| Study | Type of Intervention | Number of Cycles Pre-Operative | Number of Cycles Post-Operative | AAOMS Staging of Disease |
|---|---|---|---|---|
| Shimura et al. 2006 [41] | HBO + clarithromycin and levofloxacin followed by dexamethasone at 20 mg daily for 4 days. | N/A | N/A | Stage 2 |
| Lee et al. 2007 [42] | Sequestrectomy + PRP | Nº1 case 20 HBO at 2.4 atmospheres pressure for 90 min The other case no HBO (due to medical condition) Nº 2 case NR | Nº1 10 HBO at the conclusion of surgical treatment The other case no HBO (due to medical condition) Nº 2 case NR | NR |
| Freiberger et al. 2007 [43] | 11 × Debridement 1 × Resection 4 × just HBO | Unclear | Unclear | NR |
| Lee et al. 2011 [44] | Unclear | NR | NR | Stage 0 × 4 Stage I × 1 Stage 2 × 5 Stage 3 × 3 |
| Freiberger et al. 2012 [45] | StC (surgery + ABX + MW) × 21 (Control Group) HBO + StC (surgery + ABX + MW) × 25 (Study Group) | 0 | 40 | NR |
| Fatema et al. 2015 [46] | Drug Holiday + ABX + surgery | 20 | 10 | Stage 2 |

**Table 10.** HBO analysis of the MRONJ status at the end of follow-up; not reported (NR), poor patient compliance (PPC), computer tomography (CT). * RCT Freiberger et al. 2012 [45] showed no statistically significant improvement in cure rate compared to placebo but there was improvement in secondary outcome.

| Study | Follow-up Time | Type of Special Investigation Used during Patients' Follow-up | Treatment Complications during the Study | MRONJ Status after Treatment at the End of Follow up |
|---|---|---|---|---|
| Shimura et al. 2006 [41] | NR | NR | Patient developed acute otitis media (HBO interrupted and re-started) | Stable |
| Lee et al. 2007 [42] | 9 months | CT × 1 NR × 1 | - | Complete resolution |
| Freiberger et al. 2007 [43] | From 1 to 32 months | NR | PPC × 1 Multiple surgery × 2 HBO therapy more than one time × 4 | Remission 8 (50%) Stable 2 (12.5%) Progression 6 (37.5) |
| Lee et al. 2011 [44] | NR | NR | NR | NR |
| Freiberger et al. 2012 [45] | 24 months | NR | Fatality × 3 (after 3 months); HBO declined × 1; Immediate Crossover to HBO Group × 2; Late Crossover to HBO × 3 | 17 of 25 (68%) improved * |
| Fatema et al. 2015 [46] | NR | NR | - | Complete resolution |

*7.3. OT Study Analysis*

For OT, there were just 7 studies for analysis. Study types were described as case reports (n = 1) and case series (n = 6). All these studies were published from 2006 to 2014 (Table 11).

**Table 11.** Studies included in the OT analysis, including number of patients treated and evidence level. Case series (CS); Case report (CR); Levels of Evidence for Prognostic Studies Adapted from the American Society of Plastic Surgeons (https://www.plasticsurgery.org/Documents/medical-professionals/health-policy/evidence-practice/ASPS-Rating-Scale-March-2011.pdf).

| Author(s) | Type of Study | Total Number of Patients | Level of Evidence |
|---|---|---|---|
| Agrillo et al. 2006 [47] | CS | 30 | Level 4 |
| Agrillo et al. 2007 [48] | CS | 33 | Level 4 |
| Ripamonti et al. 2011 [49] | CS | 10 | Level 4 |
| Agrillo et al. 2012 [50] | CS | 131 | Level 4 |
| Ripamonti et al. 2012 [51] | CS | 24 | Level 4 |
| Brakus et al. 2013 [52] | CR | 1 | Level 5 |
| Brozoski et al. 2014 [53] | CS | 2 | Level 4 |

A total of 231 patients were reported, 124 female (53.67%) and 74 male (32.03%), with 14.28% of patients (n = 33) of unreported gender. Patients were treated using different OT protocols. The pooled mean of age of these patients was 60.7 years. The most common indications for antiresorptive treatment was multiple myeloma (35.93%), breast cancer (27.70%), prostate cancer (6.49%), lung cancer (6.49), renal cancer (3.46%), uterine cancer (0.43%), non-Hodgkin lymphoma (0.43%) and osteoporosis with lymphoma and thyroid cancer (4.76%). In a significant number of patients (14.28%), the drug therapy indication was not reported (Tables 12 and 13).

The most common site for MRONJ was the mandible (43.72%), followed by the maxilla (22.94%). In 8.65% of cases, MRONJ was reported in both, and the site was not reported in 24.67% of cases (Table 12).

The drug most commonly responsible for MRONJ was Zoledronate (14.71%). However, only 37 patients out of 231 had details of the specific MRONJ-associated drug reported (16.01%). OT was most commonly used as a neoadjuvant and/or adjuvant therapy, followed by surgery, in 6 studies out of 7 (Table 14).

The patients were followed for a period of time ranging from 7 to 36 months. At the end of the follow up, MRONJ was completely resolved in 44.58% of patients (n = 103), whilst 22.94% showed some improvement or remission of the disease (n = 53). No progression of the disease has been reported in any studies (Table 15); however, for 30 patients (12.98%), this data was not available.

**Table 12.** OT Preoperative epidemiologic analysis (age, sex, predisposing factors, and site of the necrosis involved). Male (M); female (F); not reported (NR).

| Study | Age/Sex | MRONJ Aetiology | Site of Necrosis |
|---|---|---|---|
| Agrillo et al. 2006 [47] | 10 M and 20 F, age ranging from 46 to 79 years old (mean age was 63 years) | Unclear | 7 patients (23.3%) in the maxilla; 18 patients (60%) in the mandible; 5 patients presented with exposed necrotic bone in both maxilla and mandible (16.7%). |
| Agrillo et al. 2007 [48] | Unclear | Unclear | Unclear |
| Ripamonti et al. 2011 [49] | 2 M and 8 F, age ranging from 53 to 77 years old (mean age 65 years old) | 8 patients after extraction; 2 from prosthetic dentures | 9 in the mandible and 1 in the maxilla |
| Agrillo et al. 2012 [50] | 49 M and 82 F, age ranging from 38 and 82 years old (mean age was 60 years old). | 70 (52%) came after dental extraction with exposure of necrotic bone in the same area. 36 (27.5%) spontaneous exposure of alveolar bone. 25 patients NR | 43 patients (33%) in the maxilla; 73 patients (55.2%) in the mandible; 15 patients presented with exposed necrotic bone in both maxilla and mandible (11.8%) |
| Ripamonti et al. 2012 [51] | 12 M and 12 F, age ranging 41–80 years old (mean age 62.5 years old) | NR | NR |
| Brakus et al. 2013 [52] | 1 F, 68-year-old | Dental extraction | Maxilla |
| Brozoski et al. 2014 [53] | 1 M, 68-year-old; 1 F, 62-year-old | 2 patients, dental extraction | 1 Maxilla; 1 Mandible |

**Table 13.** Preoperative pharmacological analysis: type of drugs, indication for drug therapy and time of drug exposure. Zoledronate acid (ZOL); Pamidornate (PAM); Alendronate (ALD); Risedronate (RES); Osteoporosis (OP); Multiple Myeloma (MM); Breast Cancer (BC); Prostate Cancer (PC); * Unclear—the number expressed do not match the actual sample analysed; Not reported (NR); Non-Hodgkin Lymphoma (nHL); Lung Cancer (LC); Uterine Cancer (UC); Renal cancer (RC); Thyroid cancer (TC).

| Study | Type of Drug | Indication for Drug Therapy | Time of Drug Exposure |
|---|---|---|---|
| Agrillo et al. 2006 [47] | NR | 23 × MM<br>5 × BC<br>1 × UC<br>1 × PC | NR |
| Agrillo et al. 2007 [48] | NR | NR | NR |
| Ripamonti et al. 2011 [49] | 9 × ZOL<br>1 × PAM | 6 × BC<br>2 × MM<br>1 × PC<br>1 × nHL | NR |
| Agrillo et al. 2012 [50] | NR | 56 × MM<br>41 × BC<br>11 × LC<br>8 × PC<br>8 × RC<br>7 × OP, Lymphoma and TC | NR |
| Ripamonti et al. 2012 [51] | 24 × ZOL | 11 × BC<br>4 × PC<br>4 × LC<br>3 × OP<br>2 × MM | Ranging from 10 to 18 months |
| Brakus et al. 2013 [52] | PAM | BC | 42 months |
| Brozoski et al. 2014 [53] | 1 × ZOL<br>1 × ALD + RES | 1 × PC<br>1 × OP | 21 months ZOL; 48 months ALD + 24 months RES. |

**Table 14.** Ozone therapy (OT) operative analysis including type of intervention and the stage of the disease. Spontaneous expulsion of necrotic bone (SENB); antibiotics (ABX); American Association of Oral and Maxillofacial Surgery (AAOMS); not reported (NR).

| Study | Type of Intervention | Number of Cycles Pre-Operative | Number of Cycles Post-Operative | AAOMS Staging of Disease |
|---|---|---|---|---|
| Agrillo et al. 2006 [47] | Surgical debridement + ABX | Unclear | Unclear | Stage 1 |
| Agrillo et al. 2007 [48] | Surgical debridement | 1 (8 session of 3 min); OT + ABX + antifungal | 1 (8 session of 3 min); OT + ABX + antifungal + Vitamin C | NR |
| Ripamonti et al. 2011 [49] | ABX + ultrasonic scaling | 3 to10 applications of OT oil (mean applications were 5.5) | None | NR (Weitzman et al. 2007 staging utilised) |
| Agrillo et al. 2012 [50] | Curettage or sequestrectomy | None | Unclear | NR |
| Ripamonti et al. 2012 [51] | ABX followed by SENB and/or sequestrectomy | 3 to 38 applications of OT gas (mean applications were 11) | None | No AAOMS used (Weitzman et al. 2007) |
| Brakus et al. 2013 [52] | Debridement + ABX (initially); eventually radical surgery | 7 applications | None | Stage 2 |
| Brozoski et al. 2014 [53] | Mouthwash + Surgical debridement | Unclear | None | Stage 2 × 1<br>Stage NR × 1 |

**Table 15.** OT analysis of the MRONJ status at the end of follow-up. Orthopantomograph (OPG); not reported (NR).

| Study | Follow-up Time | Type of Special Investigation Used during Patient's Follow-Up | Treatment Complications during the Study | MRONJ Status after Treatment at the End of Follow-Up |
|---|---|---|---|---|
| Agrillo et al. 2006 [47] | NR | NR | NR | NR |
| Agrillo et al. 2007 [48] | 7 months | NR | NR | 18 patients (54%) completed healing of the lesion; 10 patients (30%) experienced reduction of the lesion dimension; 5 patients (16%) showed no clinically relevant improvement outcomes on the lesion. |
| Ripamonti et al. 2011 [49] | 8 months | NR | None | All patients have shown sign of healing of the lesion. |
| Agrillo et al. 2012 [50] | Unclear | NR | 37 patients withdrawn from the research | 57 patients had complete resolution (60%); 28 patients had reduction of the dimension of lesions (30%); 9 patients (10%) showed no clinically relevant improvement outcomes on the lesion. |
| Ripamonti et al. 2012 [51] | Range 12 to 36 months (mean 18 months) | NR | 7 patients interrupted the treatment with OT for disease progression and 1 for fear of an experimental therapy. | 16 patients have shown complete resolution of the MRONJ. |
| Brakus et al. 2013 [52] | Unclear | Unclear | None | Reduction of the dimension of lesions. |
| Brozoski et al. 2014 [53] | 36 months × 1 18 months × 1 | OPG | None | Complete resolution. |



## 8. Risk of Bias and Review Quality Assessment

In all three case report studies, we identified a lack of clarity in many of the thirteen domains of the CARE Checklist, with missing information. We found that the lack of clarity was predominantly on follow-up and diagnostic procedures at the time of follow-up. Hence, we concluded the level of bias to be high for all the included case reports. In the nine case series studies, we reported a consistent lack of clarity in some of the seven domains of the Modified Delphi Checklist. These were predominantly regarding the outcome measurement methods, hence we considered the level of bias to be high for all case series studies. The only eligible RCT was an evaluation of adjuvant HBO therapy for people undergoing surgery. The authors of the study did not mention the generation of randomisation sequences but reported the concealment of allocation using a series of opaque envelopes containing the assignment, and we therefore rated the level of risk as unclear. The personnel involved in the study were not blinded because this was deemed to be impractical. The loss of patients to follow-up was substantial, and although a clear description of losses and withdrawals was given, data analysis was performed as-treated. Moreover, the study had a very high and unbalanced rate of crossovers between study arms. Therefore, we considered the level of risk of bias to be high across the study [41–53].

## 9. Discussion

Antiresorptive drugs are known to improve the quality of life for patients affected by bone metastasis, osteoporosis, osteopenia and Paget disease. Moreover, the new antiangiogenic drugs have been shown to be effective treatment modalities for a number of cancers. Unfortunately, increased use of these drugs has also increased the numbers of patients developing MRONJ. The risk appears to be highest in patients who require intravenous drug administration or an intake period greater than two years [12,54,55]. Moreover, the literature suggests that local and systemic factors (such as periodontal disease and diabetes) might act as predisposing factors in developing MRONJ. Although no gold standard is currently available for the treatment of MRONJ, a number of studies debate which MRONJ stage benefits the most from surgical or conservative therapy [15,56]. In general, the common opinion appears to be that for early stages of the disease (stage 0 or I), conservative management might be sufficient [6,54].

The purpose of this systematic review was to analyse the current evidence related to the treatment of MRONJ when using oxygen therapy (OT and HBO). Our findings indicate that the oxygen therapy has been used as a neoadjuvant or adjuvant therapy and may represent a viable complementary treatment or an alternative in advanced stage disease (stage II and III) where patients are unfit for aggressive surgery such as jaw resections or microvascular reconstruction. Despite the majority of studies reviewed presenting low quality evidence with a high risk of bias, there is some evidence to show the total resolution of MRONJ in 44.58% of patients with OT and 5.17% of patients with HBO, although this could equally have happened without these interventions. In addition, in a number of OT studies, the spontaneous sequestration of necrotic bone was followed by a spontaneous expulsion [47–53]. The only reported cases of MRONJ progression were amongst patients treated with HBO (10.34%, n = 6). Unfortunately, these outcomes were not reported for 13.73% (n = 43), which could represent further cases of disease progression.

This review also found that in the majority of the MRONJ cases, antiresorptive drugs were explicitly discontinued if deemed safe from the oncological point of view [41–53]. However, it is unclear if the discontinuation strategy leads to a better surgical outcome due to the long skeletal life of some antiresorptive drugs.

Amongst all the studies, we have found several perplexities which have had an impact in the quality of the research. Indeed, we have noticed that in 11 articles out of 13, there was no mention of any specific investigations during the patients follow up. Only Lee et al. [42] and Brozoski et al. [53] have reported, during the follow up, that patients had either a CT or an orthopantomograph (OPG) to assess the radiological aspect of the MRONJ lesions. Moreover, other important data were missing

in many of the articles, such as cumulative drug dose prior to developing MRONJ and patients' predisposing factors such as other medical conditions and dental history.

Only one article utilised a QoL questionnaire to assess baseline and post treatment levels at six months. The questionnaire evaluates six health domains (physical, mental, social, general, perceived health and self-esteem) and four dysfunction measurements (anxiety, depression, pain and disability), reporting no statistically significant differences between the two groups [45].

It is understandable that due to the limited incidence of MRONJ, it is difficult to improve the quality of evidence unless a common effort is applied. Therefore, the authors believe that additional high-level-evidence studies, such as multi-centre studies, case-controlled studies or randomised controlled trials, are necessary to support the efficacy and the success of oxygen therapy in managing MRONJ.

The authors advocate, in general, that the following rules should be applied for MRONJ treatment research protocols:

- Diagnosis and staging of the disease should be assessed with standardised reproducible scales and should be calibrated amongst the clinicians involved in the study.
- If randomisation is feasible, it should be carried out and described in sufficient detail to allow an assessment of whether it produced comparable groups.
- Common, quantifiable and clinically relevant endpoints (time to complete wound healing, pain, specific investigations, treatment acceptability and participant satisfaction) should be described in a sufficiently detailed manner.
- A long follow-up period of at least six months is essential if treatment effects on indolent, often long-standing MRONJ sites, are to be detected.

## 10. Conclusions

MRONJ is becoming an increasingly significant iatrogenic complication for patients undergoing antiresorptive and antiangiogenic drug therapy as these important medications reduce the morbidity and mortality rates associated with the primary disease. Unfortunately, the management of MRONJ remains controversial. This is the first systematic review of oxygen therapy for the management of MRONJ and highlights the absence of high-level evidence in the literature. Therefore, it is currently difficult to suggest OT is better or worse than HBO, or whether it is better than a placebo. In the absence of effective MRONJ treatment strategies, the available data does however suggest that further well-designed clinical studies are warranted to improve the evidence base for both OT and HBO in the management of MRONJ.

**Author Contributions:** Roberto Sacco, Racheal Leeson and Anand Lalli conceived the presented systematic review idea and the design of the study; Roberto Sacco, Joseph Nissan, Sergio Olate, Carlos Henrique Bettoni Cruz de Castro, and Alessandro Acocella contributed to the acquisition, analysis, and the interpretation of data for the work; Roberto Sacco, Racheal Leeson, Joseph Nissan, Sergio Olate, Carlos Henrique Bettoni Cruz de Castro, Alessandro Acocella and Anand Lalli drafted the paper, revised it critically and finally approved the version to be published.

**Funding:** This research received no external funding.

**Acknowledgments:** We would like to thank Professor Stefano Fedele (Oral Medicine Department, Eastman Dental Institute, London UK) for the comments that greatly improved the manuscript.

**Conflicts of Interest:** The authors declare no conflict of interest.

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
