# Peer review of "A Systematic Review of Oxygen Therapy for the Management of Medication-Related Osteonecrosis of the Jaw (MRONJ)"

_applsci, doi:10.3390/app9051026_

Round 1

Reviewer 1 Report

The authors present a systematic review of hyperbaric and ozone therapy for MRONJ. Whilst the methodology is sound, the primary limitation is the poor quality of the literature, thus making it difficult to draw valid conclusions. In particular, there was only one RCT whereby HBO was used as a adjuvant therapy vs placebo -  this study was not statistically significant for its primary outcome. All the remaining studies were either case control or case series and did not have controls to ascertain the rate of spontaneous resolution (which can be high in low stage MRONJ). There are also no head-to-head data between HBO and ozone therapy. Thus, it is difficult to draw the conclusions suggesting ozone therapy may be better than HBO, or that either therapy is better than placebo. Whilst the limitations of the reviewed data are appreciated, I would suggest that the conclusion highlight these limitations and the shortcomings of the underlying data rather than suggesting that the data are promising.

Specific comments:

Line 20/48 - MRONJ is not necessarily life changing or serious as most cases are mild and heal spontaneously

Table 1: risedronate is missing from this table

Line 73: note that Stage 0 remains contentious with some newer guidelines exclusion this classification

Line 84: the importance of dental hygiene/oral health as a major risk factor is under-appreciated

Table VII: 'triggering' not 'trigging'

Table VIII: Freiberger et al - 6 BP - which bisphosphonate did these patients receive (or was it not stated); also: risedronate rather residronate

Line 233: it states that the most common site was the mandible 18.29% followed by the maxilla 3.65%. Where were the other 80% of lesions?

Table X: results of Freiberger - not statistically significant compared with placebo in terms of cure (though some improvement in secondary outcomes). Important to state this.

Author Response

We agreed with rewording the conclusion to show the limitations of the shortcomings of the underlying data. Specifically, the word “promising” has been removed from the abstract, discussion and conclusion of the manuscript.

Line 20/48 we have made the statement less definitive in agreement with the reviewer.

Line 73 stage 0 is still presented in the AAOMS 2014 and widely used in the literature despite its acknowledged controversy.

Line 84 whilst we agree with reviewer’s comment, this risk factor was not mentioned in the articles included in this manuscript.

Line 233 as stated in the manuscript on the following line, unfortunately the majority of studies did not report the site of MRONJ.

Table I this has been added for completeness.

Table VII adjusted as suggested

Table VIII unfortunately the specific BP was not stated in this manuscript. We have clarified this in the table.

Table X thanks this has been agreed and has been added in table legend.

Reviewer 2 Report

Finally a comprehensive paper on the possible uses of ozone  in case MRNOJ. We hope that this excellent review could help and push physicians to design more complete and accurate clinical studies on the topic in the aim to reach the best evidence and protocols for ozone treatment.

Row 67 open " not followed by close "

Row 97 I suppose L-arginine and not argenine….

Row 322 I suggest to avoid to use the term alternative.. better integrative or complementary;

Row 324 Better " a feasible treatment" --- cancel alternative

Author Response

Thanks for your comments. We have made all the adjustments as suggested.

Reviewer 3 Report

It would have been interesting to know methods of administration and dosages of HBO vs OT to select the most effective treatments.

Author Response

Thanks this is a valuable point. However, this information is lacking in the manuscripts included in this systematic review.

Round 2

Reviewer 1 Report

The authors present an updated manuscript with minor revisions based on previous comments and observations. In particular, minor revisions that directly address previous suggestions have been made. However, this does not change the fact that the data remains poor and it is difficult to draw conclusions from the systematic review. Whilst the abstract conclusion has been appropriately modified in the abstract, this conclusion does not match the manuscript conclusion (line 356 or 376) which does not draw a conclusion other than suggesting what a well-designed study may look like (would be better placed in the discussion). Also, the discussion (lines 327-333) still suggest that HBO and ozone therapy may be beneficial despite the lack of data.

Minor edits to language/spelling are still required: examples included but not limited to "trigging" still appears in some tables, and 'complimentary' should be 'complementary' (line 330)

Author Response

Dear Editor,

thank you for the opportunity to respond to the reviewer`s comments on our manuscript. We have made revisions and offered rebuttals where appropriate as outlined below. The manuscript with ˜track changes' enabled to show these changes. Overall, this has improved our manuscript and we thank the reviewers for their comments.

As suggested, we have moved the important points about future study designs into the discussion and conclusions have been rewritten to clarify that the available data for oxygen therapy is poor and also to highlight the context of the lack of alternative effective management strategies for MRONJ. Editing for English spelling and grammar has been redone as suggested.

Roberto Sacco

Round 3

Reviewer 1 Report

The report discussion and conclusion now more accurately represent the low level evidence and appropriate conclusions that can be drawn from these.